# OpenReview forum: "Text Speaks Louder than Vision: ASCII Art Reveals Textual Biases in Vision-Language Models"
_colmweb.org/COLM/2025/Conference — COLM 2025_

### Official Review · Reviewer_Fq3A · 2025-05-08

**Rating:** 7
**Confidence:** 4
**Ethics Flag:** 1

**Summary:**

This paper investigates how vision-language models (VLMs) process ASCII art, where textual elements collectively form visual patterns. The authors create a dataset of adversarial ASCII art images, where character-level semantics deliberately contradict global visual patterns. Experiments show that VLMs consistently prioritize textual information over visual patterns and this bias cannot be effectively mitigated via prompt engineering strategies, providing important guidance for future model development.

Overall, the paper is well-written. The proposed method is interesting, effectively reveals a strong text-priority bias in VLMs, and provides insights for future model development.

**Reasons To Accept:**

- The paper reveals a strong text-priority bias in VLMs via adversarial ASCII art images, providing important guidance for future model development.

- The paper proposes several prompt engineering strategies to mitigate the text-priority bias.

- The method of using ASCII art to reveal biases in VLMs is interesting and could be insightful for revealing and mitigating other potential biases in  VLMs in the future.

- The paper is well-written and easy to follow.

**Reasons To Reject:**

- Line 57: The claim that the experiments underscore "the need for architectural changes" may be overstated. The paper only evaluates prompt-based mitigation strategies, without exploring more direct approaches such as fine-tuning VLMs. Therefore, the conclusion advocating for architectural changes appears not well-supported.

- Minor issues:
  - Figure 5: It is unclear what a visual hint is and how it is provided to VLMs.
  - The title of Section 2.2: VLM and language bias -> VLM and Language Bias

---

> ### Author Response · Authors · 2025-06-02
> **Response to Reviewer Fq3A**
>
> We sincerely thank the reviewer for the positive evaluation and constructive feedback. We are encouraged that you found our research "interesting and insightful" with valuable methodology for future work. Below, we address each of your comments with detailed responses.
>
> ### **1. Regarding "Architectural Changes" Conclusion Statement**
>
> Thank you for highlighting this important expression issue. We will revise the original sentence to ensure more accurate and balanced wording. Specifically, we will change the phrase *"underscore the need for architectural changes"* to *"suggest the need for deeper analysis and more comprehensive solutions."*
> This revised phrasing better reflects our findings while avoiding overstatement about specific technical requirements.
>
> ### **2. Response to Minor Issues**
>
> We thank you for this helpful suggestion.
> - For **Figure 5**, we provided more detailed examples in Appendix C, and we will revise the figure with clearer examples and add clearer explanations in the caption.
> - For the **title of Section 2.2**, we will correct "VLM and language bias" to "VLM and Language Bias" to maintain consistent title formatting.
>
> Thanks again for your thorough and constructive review. We believe these modifications will significantly improve the paper's rigor and clarity.

---

> > ### Comment · Reviewer_Fq3A · 2025-06-05
> >
> > Thank you for your response. I will retain my positive evaluation of the paper.

---

> > > ### Author Response · Authors · 2025-06-06
> > >
> > > We sincerely appreciate your recognition of our work’s value and will further improve it based on your suggestions.

---

### Official Review · Reviewer_ygS7 · 2025-05-13

**Rating:** 5
**Confidence:** 3
**Ethics Flag:** 1

**Summary:**

This paper explores a very interesting problem on VLMs about how to process the ASCII art. It mainly analyzed the case that textual and visual information conflict. This paper proposed a data that includes ASCII art images constructed from diverse character sets for testing how textual information affects visual understanding. This paper reveals a finding that VLMs prioritize textual information over visual patterns. It is expected that the VLMs cannot handle the ASCII art inputs because of lack of ASCII art training data in most of VLMs. In general, this paper worked on a special and interesting ”text + image” ASCII art emotion detection task.

**Questions To Authors:**

Please check the "reasons to reject".

**Reasons To Accept:**

•	This paper proposed an ASCII art data and evaluate it on five state-of-the-art VLMs. This data includes various character types with different semantic complexities. This data will be a very useful data for the community.

•	The analysis of this paper reveals a bias when LLM processes ASCII art: model will extract more textual information than visual. It will become more severe when character are semantically complex. Although this paper explored few ways to handle this, the problem cannot be solved well.

**Reasons To Reject:**

•	The exploration of this paper is only on one special emotion detection task. There are two concerns: 1) the analysis on one task is too limited. The conclusions maybe be not a general one for other tasks. 2) This paper hasn’t fully explored the ability to understand the textual information contained in the ASCII art image. More deep analysis is needed to evaluate the performance of tasks that will use the textual information from input.

•	The bad performance on handling this type of input is expected. This paper should go deeper and explore more ways like finetuning a model to handle this new task.

---

> ### Author Response · Authors · 2025-06-02
> **Response to Reviewer ygS7**
>
> We thank the reviewer for the positive feedback and constructive suggestions. We are encouraged by your recognition of our dataset’s value and the importance of our findings on text-priority bias. We also appreciate your ideas on expanding tasks and exploring mitigation. Below, we provide clarifications and new results in response to your comments.
>
> ### **1. Task Scope and Generalizability Issues**
>
> We chose sentiment detection for several practical reasons. First, it has broad real-world applications in content moderation, social media monitoring, and human-computer interaction. Second, it provides clear ground truth labels that make controlled evaluation reliable. Most importantly, adversarial ASCII art is already being used in online platforms. This makes our research both timely and practically relevant.
>
> Although our focus is on sentiment, our findings apply more broadly. The core contribution is understanding how visual elements can contradict textual content. This affects virtually all text-visual multimodal tasks. Our L2–L7 scale captures a range of text-symbol patterns found in real-world scenarios, from simple characters to complex words and poetic phrases. These character types cover most cases that most tasks may encounter, giving our framework strong generalizability.
>
> The main issue we study is text-bias, which is a key challenge affecting VLM performance and development. Our analysis provides targeted insights into how this bias interacts with adversarial ASCII art.
>
> We also thank the reviewer for the thoughtful suggestions, which encourage us to extend our evaluation in future work to tasks such as document classification, visual question answering, and image captioning.
>
> ### **2. In-depth Analysis of Text Information Understanding Capabilities**
>
> We thank the reviewer for raising this important discussion. We actually deeply explored VLMs' capabilities in understanding textual information in ASCII art:
>
> Our L1–L7 design systematically tests how VLMs handle text with different levels of semantic complexity. Instead of mixing all text types together, we provide a fine-grained categorization, which reveals clear and detailed performance patterns:
>
> - **L1–L2**: Non-semantic symbols → 100% accuracy
> - **L3–L4**: Medium semantic complexity → Significant accuracy decline
> - **L5–L7**: High semantic complexity → Near 0% accuracy
>
> Our results show that the issue is not that VLMs fail to understand text, but that they over-rely on it while ignoring visual information. Different types of text cause different levels of interference with visual perception. This text bias is the core problem our study aims to reveal.
>
> ### **3. Regarding "Expected Results" and Solution Exploration**
>
> We sincerely thank the reviewer for this insightful comment. We appreciate the opportunity to clarify our research contributions and address concerns about solution exploration.
>
> While poor VLM performance on ASCII art may be expected, our study still reveals important insights. We found not just a simple good/bad outcome, but a clear gradient linked to semantic complexity—from 100% accuracy at L1 to 0% at L5 and L7, with L6 (emoji) showing unexpectedly high accuracy (88–92%).
>
> In Sections 4.2 and 4.3, we explored several promising approaches, including visual enhancement techniques. Their limited success further underscores the seriousness of the issue.
>
> We fully agree that fine-tuning is a promising strategy. However, the open-source models we tested performed poorly (Qwen averaged 25%, LLaMA 7%, while GPT-4o reached 50%), which made it difficult to build a reliable baseline for meaningful downstream comparison.  Thanks for the insightful suggestions. Due to the tight review timeline and limited computational resources, we were unfortunately unable to conduct full fine-tuning. Thanks for the insightful suggestions.
> We consider this an important direction for future work. We plan to explore whether enhancing the model’s spatial reasoning  can improve its ability to handle ASCII art, and extend our study to structured, text-heavy formats such as tables and technical documents. Thanks again for the valuable suggestion.

---

> ### Author Response · Authors · 2025-06-08
>
> Dear Reviewer ygS7:
>
> Thank you for your helpful comments. We would like to kindly invite you to take a look at our rebuttal. We hope our responses clarify your concerns, and we are happy to follow up on any further questions.

---

### Official Review · Reviewer_3Tgy · 2025-05-13

**Rating:** 5
**Confidence:** 3
**Ethics Flag:** 1

**Summary:**

This study analyzes how vision-language models (VLMs) respond to adversarial attacks involving ASCII art, specifically focusing on the relationship between visual patterns and the characters used to compose the artwork. The research compares ASCII art representing negative semantic targets using characters with varying levels of neutrality or positivity, examining how the type of character influences the success of the attack. In addition, it investigates the robustness of VLMs to visual factors such as font size and inter-character spacing. The proposed analysis provides a comprehensive examination of ASCII art-based adversarial attacks and offers insights into potential vulnerabilities of VLMs.

**Questions To Authors:**

* Beyond the suggested combination of positive words and negative characters, the study could explore a broader variety of attacks, such as using characters from different languages or creating negative imagery unrelated to text. This would further enrich the analysis and enhance the relevance of the study.

* Please provide further explanation regarding the analysis related to the L6 class.

**Reasons To Accept:**

* The paper addresses a highly intriguing problem: adversarial attacks using ASCII art.

* The authors design a tightly controlled experimental setting to enable step-by-step analysis of VLM behavior in response to complex ASCII art attacks. Specifically, the target words are negative, while the characters used to depict them are neutral or positive, allowing a clear separation of visual pattern effects from those of character-level semantics.

* The study outlines several practical considerations for defending against ASCII art-based adversarial attacks.

**Reasons To Reject:**

* The behavior of VLMs under ASCII art-based adversarial attacks is likely to be more complex than suggested in this work, which appears somewhat narrow in scope. This limitation may stem from the difficulty in controlling commercial models such as GPT-4o, which include additional features like OCR modules or moderation filters.

* The study collects data only for images representing negative words. This raises concerns about class bias, as the evaluation does not control for potential asymmetries between positive and negative classes. It would be beneficial to also include cases that use negative characters to depict positive words. Such cases could still allow a clean separation between visual pattern and character-level semantics, and may even reveal potential vulnerabilities in visual-feature-based moderation systems.

* The practical impact of the attack seems somewhat limited. It would strengthen the paper to provide additional evidence or argumentation about the potential harm caused by such adversarial ASCII art examples.

---

> ### Author Response · Authors · 2025-06-02
> **Response to Reviewer 3Tgy**
>
> We sincerely thank the reviewer for the recognition of our work and the valuable feedback provided. We greatly appreciate your constructive suggestions regarding experimental scope, class balance considerations, and practical impact assessment. Below, we provide detailed clarifications and new experimental results to address each point raised.
>
> ### **1. VLM Behavioral Complexity and Research Scope**
>
> Thank you for raising this important point.  We agree that VLM behaviour under ASCII-art attacks is rich and multifaceted. We position our work as a focused first step: we isolate one clean, well-controlled attack family to expose core failure patterns.
>
> We extensively tested multiple SOTA models, including white-box models (LLaMA and Qwen) alongside commercial systems. As shown in Table 2, all models made the same classification mistakes, even though they have different architectures and sizes. This consistent behavior supports the validity of our findings.
>
> Additionally, we did not choose white-box models as our main test models because they performed poorly on our task (Qwen averaged 25%, LLaMA 7%, while GPT-4o reached 50%). Their low accuracy made it hard to establish meaningful baselines, which would weaken the value of later analysis.
>
> We also agree that hidden OCR or moderation modules complicate control over closed-source models. We therefore conducted detailed analysis of response patterns between our baseline and *No OCR* groups, revealing behavioral differences, particularly in L5 and L7 complexity levels:
>
> | Complexity Levels | Positive Responses (baseline → No OCR) | Neutral Responses (baseline → No OCR) |
> |:------------------|:----------------------------------------:|:---------------------------------------:|
> | L5                | 80% → 50% (-30%)                         | 20% → 50%                               |
> | L7                | 84% → 6%  (-78%)                         | 16% → 94%                               |
>
> These changes demonstrate that our OCR prompt was effective (to some extent); the model followed our instructions and adjusted its behavior. Under *No OCR* conditions, the model relied less on textual content and attempted to focus on visual layout patterns. However, the model still struggled to correctly interpret the visual structure, with most responses indicating that the image "lacks clear imagery to convey a strong emotional tone".
>
> Thanks again for the valuable suggestion, we will include more details and a discussion of this limitation in the revised version.
>
>
> ### **2. Rationality of Dataset Design**
>
> We appreciate the reviewer raising this important design discussion. Our dataset design reflects several deliberate choices based on empirical findings and methodological requirements:
>
> First, our work adopts an adversarial testing approach focused on probing text-bias rather than measuring overall performance. This single-label dataset design aligns with established adversarial testing practices in security and capability boundary research [1–3]. We use ASCII art images as crafted adversarial inputs (attacks) targeting specific VLM vulnerabilities in text-visual processing.
>
> Second, this mirrors genuine attack scenarios where malicious users employ positive-looking characters to create harmful content that evades detection systems. These scenarios are both highly impactful and directly aligned with our research goal, which is why we center our design on them.
>
> We thank the reviewer for suggesting “negative-char / positive-word” examples. During our early validation phase, we built a pilot set (50 samples) where negative characters were rearranged into visually positive shapes (e.g., "JOY"). We asked three independent annotators to label this pilot set. They disagreed on 35% of the cases because each picture sent two conflicting signals: (i) a strong positive initial impression from the friendly overall layout, and (ii) a negativity-bias reaction [4–6] once the harmful word was noticed. This clash of cues produced ambiguous judgments, making the labels too noisy for a reliable evaluation. So we dropped the split in the released dataset to keep evaluation reliable.
>
> We agree that dual-class adversarial art is valuable for broader safety analysis. We will add more discussion on this point in the revised version. To support future work, we will release a lightweight dataset creation script so the community can easily build and extend their own datasets. We sincerely thank the reviewer for this valuable suggestion.

---

> > ### Comment · Reviewer_3Tgy · 2025-06-09
> >
> > * Concerns about the practical impact of ASCII art-based attacks
> >
> > While I agree that this study clearly demonstrates the limitations of current VLMs, I remain unconvinced about the real-world negative impact of ASCII art-based attacks. Personally, I did not find such attacks to be harmful when I encountered them, which may explain my skepticism. To strengthen the argument, it would be helpful to provide more objective evidence showing how and to what extent ASCII art attacks have a harmful effect on users in practical scenarios.
> >
> > * Suggestions for a more rigorous analysis of cross-lingual bias
> >
> > To conduct a more thorough analysis of bias arising from cross-lingual input, I suggest controlling for the sentiment of the words and varying only the language. This would allow a cleaner assessment of cross-lingual bias. The current experimental setup in the response does not adequately isolate the effects of language and sentiment polarity, making it difficult to attribute the bias to a specific factor.
> >
> > * On the inclusion of positive cases in the dataset
> >
> > As mentioned in the author response, including positive cases in the dataset would be a welcome addition and could further enhance the comprehensiveness of the evaluation.

---

> > > ### Author Response · Authors · 2025-06-10
> > > **Response to Reviewer 3Tgy**
> > >
> > > Thank you for your response and feedback. Your comments helped us better clarify our contributions and limitations. In the following, we provide further explanations and add new experimental insights.
> > >
> > > ### On the practical impact of ASCII art-based attacks
> > >
> > > We understand your concern and appreciate your perspective. While individual exposure to ASCII art may vary, these attacks are real. We identified over 300 ASCII-based offensive content examples on emojicombos.com alone, with categories including harassment, hate speech, and inappropriate imagery. In addition, forums like Reddit and Steam have active groups that share and organize offensive ASCII art.
> > >
> > > ### Suggestions for a more rigorous analysis of cross-lingual bias
> > >
> > > We greatly appreciate this valuable suggestion and agree that “controlling for sentiment while varying only language would allow a cleaner assessment of cross-lingual bias.”
> > >
> > > We faced practical challenges during implementation. The structural complexity of Chinese characters makes it difficult to create negative Chinese words from positive ones without degrading ASCII art quality and recognizability. As a compromise, we used positive Chinese characters to construct negative English words. This approach preserves visual clarity while still enabling us to explore cross-lingual effects.
> > >
> > > To enable a more rigorous analysis, we constructed 100 negative Spanish samples using emotionally positive Spanish words (L5 level). We also used prompts written in Spanish:
> > >
> > > > ¿Cuál es la emoción de esta imagen? ¿Positiva, negativa o neutra?
> > >
> > > and tested them on GPT-4o. The results were strikingly similar to English: GPT-4o classified most images as positive, clearly relying on the emotional tone of the embedded text. This contrasts with the model’s behavior on Chinese character inputs, where Latin-based text appears more prone to triggering text bias. This may relate to cross-lingual alignment issues.
> > >
> > > | Complexity Level | Positive | Negative | Neutral |
> > > |------------------|----------|----------|---------|
> > > | L5               | 87       | 0        | 13       |
> > >
> > > Due to time constraints and our limited linguistic capabilities, we were unable to conduct more experiments across other language pairs. We will discuss our findings and limitations thoroughly in the revised version. Thank you for this valuable suggestion.
> > >
> > > ### On the inclusion of positive cases in the dataset
> > >
> > > Thank you for the suggestion. We agree that “including positive cases in the dataset would be a welcome addition and could further enhance the comprehensiveness of the evaluation.” To further explore this, we conducted an additional test using positive samples.
> > >
> > > Specifically, we selected 100 strongly positive sentiment words (e.g., HAPPY, JOY, NICE…) and applied the same method to generate a balanced L1–L7 dataset for evaluation.
> > >
> > > > Note: For L5–L7, we did not construct adversarial ASCII art. Instead, we used emotionally neutral words, emojis, and short passages to preserve the overall positive sentiment of the images. If we had used negative-looking characters, the global emotional tone would no longer be positive.
> > >
> > > Due to time and computational constraints, we focused our testing on the best-performing model, GPT-4o. The results are interesting:
> > >
> > > | Test Type                 | L1   | L2   | L3   | L4   | L5   | L6   | L7   | Avg.  |
> > > |---------------------------|------|------|------|------|------|------|------|--------|
> > > | Original Test (all negative) | 100% | 100% | 60%  | 4%   | 0%   | 88%  | 0%   | 50.3% |
> > > | New Test (all positive)      | 100% | 100% | 69%  | 0%   | 0%   | 91%  | 0%   | 51.4% |
> > >
> > > In high-complexity groups (L5, L7)—the core test cases in our diagnostic tool—there is little performance difference between positive and negative samples, and model performance remains poor overall.
> > >
> > > In low-complexity groups (L1–L3), positive samples perform slightly better (e.g., L3: 60% → 69%). One possible explanation is that models are often aligned to favor safe or positive generations.
> > >
> > > We think that model behavior remains largely consistent across positive and negative examples, especially in the high-complexity cases we care most about. Adding positive samples does not change our diagnosis of the core vulnerability.
> > >
> > > We thank the reviewer for this helpful suggestion. We will include the corresponding results and discussion in the revised version to make the paper more complete and rigorous.

---

> ### Author Response · Authors · 2025-06-02
> **Response to Reviewer 3Tgy**
>
> ### **3. Practical Impact and Potential Harm**
>
> We appreciate the reviewer's important question regarding practical impact. We would like to emphasize that our findings reveal significant real-world vulnerabilities with serious implications for VLM deployment:
>
> 1. **Limited training data creates blind spots**: Current VLM training datasets include limited ASCII art examples. This leaves models unprepared for such inputs in real-world use.
>
> 2. **Text-bias**: We find that VLMs often focus on text and ignore the surrounding visual layout. This makes them likely to miss harmful content hidden in visual patterns.
>
> 3. **VLMs are being used in content moderation**: Some video platforms are starting to rely on VLMs for content review [7]. Any weakness in these models could directly affect online safety.
>
> 4. **Attacks are easy to create and spread**: ASCII art is simple to make and share. Attackers don’t need special skills, which makes this vulnerability easy to exploit at scale. It's easy to find discussions in some online communities about insulting and offensive ASCII art
>
> In our tests, powerful GPT-4o misclassified 80% (L5) and 84% (L7) of harmful images as positive—0% accuracy. Such failure further confirms how serious the problem is. We will expand this section in the revised version with more real-world examples and additional discussion.
>
>
> ### **Q1: Exploration of Extended Attack Types**
>
> We thank the reviewer for this valuable suggestion. We fully agree that richer attack variants such as cross-lingual characters and non-textual negative imagery would deepen the analysis.
>
> Due to the tight review timeline and limited computational resources, we are unable to build and test a full new dataset. Instead, we ran a small pilot to explore cross-lingual effects. We used Chinese positive words to construct negative English words (focusing on L5, 100 samples). The result is really impressive: 94% of responses were correct (negative). The models successfully recognized the visual layout.
> | Complexity Level | Positive | Negative | Neutral |
> |:------------------|:--------:|:--------:|:--------:|
> | L5               | 0        | 94       | 6        |
>
> This suggests that GPT-4o may be more vulnerable to interference from English text. This could be due to its English-heavy training data or the visual differences between Chinese characters and Latin letters. We plan to explore this in greater depth in future work.
>
> We are grateful for the reviewer’s thoughtful feedback and will incorporate these new findings and discussions in the revised version of the paper.
>
> ### **Q2: Detailed Analysis of L6 Category**
>
> We initially expected emojis like "laughing-😀" to cause significant interference due to their semantic richness, similar to other high-complexity categories. However, all models performed unusually well on L6 (GPT-4o: 88%, Gemini: 88%, Qwen: 92%), contrasting sharply with L5 and L7 performance.
>
> A likely explanation is that VLMs treat emojis as visual patterns rather than text. Unlike LLMs that process emojis as semantic Unicode characters, VLMs may rely more on visual pathways. This is supported by longer response times (35% higher on average), suggesting additional visual processing.
>
>
> **References**
>
> [1] Wallace, E., Feng, S., Kandpal, N., Gardner, M., & Singh, S. (2019). *Universal adversarial triggers for attacking and analyzing NLP*. arXiv:1908.07125.
>
> [2] Rajpurkar, P., Jia, R., & Liang, P. (2018). *Know what you don't know: Unanswerable questions for SQuAD*. arXiv:1806.03822.
>
> [3] Chao, P., Debenedetti, E., Robey, A., Andriushchenko, M., Croce, F., Sehwag, V., ... & Wong, E. (2024). *Jailbreakbench: An open robustness benchmark for jailbreaking large language models*. arXiv:2404.01318.
>
> [4] Baumeister, R. F., Bratslavsky, E., Finkenauer, C., & Vohs, K. D. (2001). *Bad is stronger than good*. Review of General Psychology, 5(4), 323–370.
>
> [5] Öhman, A., Flykt, A., & Esteves, F. (2001). *Emotion drives attention: detecting the snake in the grass*. Journal of Experimental Psychology: General, 130(3), 466.
>
> [6] Vuilleumier, P., & Schwartz, S. (2001). *Emotional facial expressions capture attention*. Neurology, 56(2), 153–158.
>
> [7] Lu, X., Zhang, T., Meng, C., Wang, X., Wang, J., Zhang, Y., ... & Gai, K. (2025). VLM as Policy: Common-Law Content Moderation Framework for Short Video Platform. arXiv preprint arXiv:2504.14904.

---

> ### Author Response · Authors · 2025-06-08
>
> Dear Reviewer 3Tgy,
>
> We appreciate your thoughtful feedback. We have provided detailed responses and additional results in our rebuttal. We would be grateful if you could kindly take a look, and we are happy to provide further clarification if needed.

---

### Official Review · Reviewer_hQ3b · 2025-05-13

**Rating:** 6
**Confidence:** 4
**Ethics Flag:** 1

**Summary:**

This paper investigates how vision-language models (VLMs) interpret adversarial ASCII art images where textual content contradicts visual intent. The authors construct a dataset of negative-sentiment images rendered using characters with increasing semantic richness, from meaningless symbols to positive words and poems. Five widely used VLMs (e.g., GPT-4o, Claude, Gemini) are evaluated on a sentiment classification task. Results show that all models exhibit a strong text-priority bias. Specifically, as character semantics become richer, accuracy drops with models frequently misclassifying visually negative ASCII art as positive. The study explores mitigation strategies such as visual parameter tuning and prompt engineering, but finds them only marginally effective. The work highlights the modality misalignment in current VLMs and raises concerns for security risks in content moderation systems that rely on current VLMs.

**Questions To Authors:**

1. How does your method fundamentally differ from existing work [2, 3]? What are the novel insights beyond a new dataset?
2. The prompt includes the instruction like "CRITICAL INSTRUCTION: You must NOT use OCR, text recognition, or any text processing tools". However, for closed-source VLMs like GPT-4o, which may implicitly rely on text recognition during inference, how can this control be effectively enforced?
3. In the setup, the image contains embedded text as part of its visual structure. When models are asked to classify "the sentiment of the image", this instruction is ambiguous. It could refer to the sentiment conveyed by the overall visual layout, or to the sentiment of the words within the image. How to handle the control for this ambiguity?
4. There is no supplementary material attached. Will the dataset and code be released?

[1] Zhang, X., Li, S., Shi, N., Hauer, B., Wu, Z., Kondrak, G., ... & Lakshmanan, L. V. (2024). Cross-Modal Consistency in Multimodal Large Language Models. arXiv preprint arXiv:2411.09273.

[2] Berezin, S., Farahbakhsh, R., & Crespi, N. (2024). Read Over the Lines: Attacking LLMs and Toxicity Detection Systems with ASCII Art to Mask Profanity. arXiv preprint arXiv:2409.18708.

[3] Hong Wang, Xuan Luo, Weizhi Wang, and Xifeng Yan. Bot or human? detecting chatgpt imposters with a single question. arXiv preprint arXiv:2305.06424, 2023.

**Reasons To Accept:**

1. The idea of using adversarial ASCII art as a testbed is interesting. It draws a parallel to traditional image-based adversarial attacks, but replaces pixels with semantic textual units.
2. Some findings are insightful. For example, models tend to produce incorrect positive predictions when the embedded text is semantically positive. Also, lower resolution improves sentiment classification accuracy, which aligns with human perception.
3. The experimental design is thorough, covering multiple VLMs and settings, such as semantic complexity levels and image resolution, to probe robustness.
4. While a solution is not proposed, the authors explore mitigation strategies such as visual parameter tuning and prompt engineering, providing a useful starting point for future work.

**Reasons To Reject:**

1. The paper adds to an already growing number of benchmark-style evaluations of VLMs, without proposing a new modeling technique or theoretical insight.
2. Similar issues around modality inconsistency, particularly the dominance of text over image, have been previously explored [1]. The setup focuses solely on ASCII art sentiment classification, which is a narrow and synthetic domain, also limiting generalizability to real-world multimodal applications.
3. Prior work has shown similar findings regarding the limitations of VLMs when interpreting character-based or adversarial text layouts. For example, [2] explores ASCII-based attacks on toxicity detection, and [3] reports that VLMs perform poorly on tasks requiring ASCII image understanding. However, these works are not discussed, leaving the contributions of the current paper unclear.
4. The proposed dataset is label-biased, with all samples labeled as negative. Given that models are often aligned to favor safe or positive generations, this setup may artificially penalize models that behave in line with their alignment objectives. This concern is not acknowledged or controlled for in the evaluation. (A majority voting can get perfect accuracy easily.)

---

> ### Author Response · Authors · 2025-06-02
> **Response to Reviewer hQ3b**
>
> We thank the reviewer for the constructive feedback and insightful suggestions. We are encouraged that you found our work interesting and saw value in using adversarial ASCII art to study modality misalignment in VLMs. We also appreciate your recognition of our experimental design and mitigation efforts. Below, we address your questions and provide additional clarifications and results.
>
> ### 1. **Contribution and Modeling Insights**
>
> We appreciate the reviewer's inquiry regarding our contributions. While evaluative, our study goes beyond standard benchmarks and offers key insights into how VLMs behave under semantic conflict. Our key contributions are:
>
> - **First quantification of semantic complexity's progressive impact**:  We created the **L2–L7 framework** using text types with different semantic complexity. These include simple characters, numbers, common words, emojis, and short poetic phrases. Our framework shows that VLM text bias varies continuously rather than in a binary manner, providing new understanding of multimodal fusion.
>
> - **A scalable diagnostic framework for text-visual conflicts**:  We propose a systematic framework to evaluate text-visual conflicts. This goes beyond a dataset—it is a **scalable paradigm** applicable to broader multimodal scenarios. We include a wide range of character and text types that cover most patterns found in real-world tasks, making our framework generalizable to other applications.
>
> These contributions advance our understanding of fundamental limitations in current VLM architectures and provide actionable insights for future model development.
>
> ### 2. **Fundamental Differences from Existing Work**
>
> Our study introduces several key advances that distinguish it from prior research:
>
> - **Progressive Analysis**:  Unlike [1], which treats all text uniformly, we introduce L2–L7 semantic complexity levels that reveal how the impact of embedded text varies continuously. This allows a fine-grained analysis of VLM degradation patterns.
>
> - **Novel Methodology**:  While [2,3] focus on LLMs, we develop adversarial ASCII art that creates deliberate text-visual semantic conflicts specifically for **VLM evaluation**. Reference [3] did test VLMs, but lacked experimental data and analysis.
>
> These differences make our work a basic diagnostic tool for understanding VLM limitations. We will strengthen these distinctions and add more detailed comparisons in our revision.
>
> ### **3. Rationality of Dataset Design**
>
> We appreciate the reviewer raising this important design discussion. Our dataset design reflects several deliberate choices based on empirical findings and methodological requirements:
>
> - **Adversarial Testing Focus**: Our work adopts an adversarial testing approach focused on probing text-bias, rather than measuring overall performance. This single-label dataset design aligns with established adversarial testing practices in security and capability boundary research [4–6]. We use ASCII art images as crafted adversarial inputs targeting specific VLM vulnerabilities in text-visual processing.
>
> - **Realistic Attack Scenarios**: This mirrors genuine attack scenarios, where malicious users employ positive-looking characters to create harmful content that evades detection systems. These scenarios are both impactful and directly aligned with our research goal, which is why we center our design on them.
>
> - **Challenges with Positive Cases**:  Our early validation revealed fundamental challenges in creating positive adversarial examples: **Negative characters forming positive patterns** failed due to negativity bias—the negative textual elements dominated emotional interpretation [7–9], possibly overwhelming the intended positive visual impression. Even human annotators disagreed on the labels. **Positive characters forming positive patterns** removed the text-visual conflict, leading models to achieve artificially high accuracy (L5: 100%, L6: 95%, L7: 86%) by reading the text alone, rather than processing the visual layout.
>
> Given these issues, we adopted an all-negative sample design, which provides higher consistency across different ASCII-art groups and better serves our diagnostic objectives. We appreciate this feedback and will expand our discussion of dataset design rationale in the revision to better explain our choices and methodological considerations.

---

> > ### Comment · Reviewer_hQ3b · 2025-06-04
> >
> > Thanks for the response and helpful clarifications. The L2–L7 setup is a nice addition, but the paper still fits into the growing VLM benchmark work. The main ideas, text over image bias and ASCII input issues, have been shown before, though the setup here adds some new angles. It is still not convincing to use only negative labels. While it might fit the adversarial testing, a balanced dataset would help generalize the findings. If not, more discussion of what this setup reveals would be interesting. In short, the paper might work best as a diagnostic tool or an adversarial attacking one. With that framing, the contribution is clearer. I intend to stay with my current position, but with no objection to accepting this paper.

---

> > > ### Author Response · Authors · 2025-06-05
> > > **Response to Reviewer hQ3b**
> > >
> > > Thank you for your response and feedback. We sincerely appreciate your time, expertise, and careful evaluation of our work. Your comments helped us better clarify our contributions and limitations. In the following, we provide further explanations and add new experimental insights.
> > >
> > > ### **Contributions and Innovations**
> > >
> > > We see the main value of our work in its role as a “diagnostic tool”. While existing studies on text-image bias typically examine cases where text and visual features can be independently manipulated, our work explores a fundamentally different scenario: ASCII characters that simultaneously function as both semantic text and visual structure. This dual role creates what we term an *intra-modal conflict*, which poses new challenges for multimodal models and has received limited attention in prior work. Our contributions include:
> > >
> > > - **Fine-Grained Insight**:  We systematically investigate the relationship between complexity and text bias. Our experiments show that high-complexity content (L5–L7) strongly triggers text bias, while simple characters or digits (L2–L4) have a small effect. This structured finding reveals a clear link between semantic richness and model vulnerability, providing a deeper understanding of the conditions under which text bias emerges.
> > >
> > > - **Practical Implications**:  Based on our deeper understanding of the text-bias mechanism, our findings point to several important directions for further exploration:
> > >   - Digits and symbols trigger minimal text bias. This raises an interesting question — are mathematical expressions less affected by text-driven interference?
> > >   - Can we reduce text bias by selectively masking high-semantic keywords in the input, prompting the model to rely more on visual cues?
> > >
> > > These perspectives have not been addressed in prior work. They open up new opportunities for designing more robust VLM pipelines and improving performance in safety-critical applications.
> > >
> > > ### **2. Only Negative Labels**
> > >
> > > Thank you for the suggestion. We agree that “a balanced dataset would help generalize the findings” is an important point. To further explore this, we conducted an additional test using positive samples.
> > >
> > > Specifically, we selected 100 strongly positive sentiment words (e.g., HAPPY, JOY, NICE…) and applied the same method to generate a balanced L1–L7 dataset for evaluation.
> > > **Important note**: For L5–L7, we did not construct adversarial ASCII art. Instead, we used emotionally neutral words, emojis, and short passages to preserve the overall positive sentiment of the images. If we had used negative-looking characters, the global emotional tone would no longer be positive.
> > >
> > > Due to time and computational constraints, we focused our testing on the best-performing model, GPT-4o. The results are interesting:
> > >
> > > | Test Type         | L1   | L2   | L3   | L4   | L5   | L6   | L7   | Avg.  |
> > > |:------------------|:----:|:----:|:----:|:----:|:----:|:----:|:----:|:-----:|
> > > | Original Test (all negative) | 100% | 100% | 60%  | 4%   | 0%   | 88%  | 0%   | 50.3% |
> > > | New Test (all positive)      | 100% | 100% | 69%  | 0%   | 0%   | 91%  | 0%   | 51.4% |
> > >
> > > In high-complexity groups (L5, L7)—the core test cases in our diagnostic tool—there is little performance difference between positive and negative samples, and model performance remains poor overall.
> > >
> > > In low-complexity groups (L1–L3), positive samples perform slightly better (e.g., L3: 60% → 69%). This may relate to the reviewer’s point that "models are often aligned to favor safe or positive generations."
> > >
> > > However, we think that model behavior remains largely consistent across positive and negative examples, especially in the high-complexity cases we care most about. Adding positive samples does not change our diagnosis of the core vulnerability.
> > >
> > > We are grateful for your constructive input, which has strengthened the depth and clarity of our study. The revised version will include the additional results and discussion accordingly.

---

> > > > ### Comment · Reviewer_hQ3b · 2025-06-05
> > > >
> > > > Thanks for the additional information, especially for confirming the assumption about using positive labels. I have updated my rating based on this clarification, improved understanding, and the authors' commitment to revising the next version.

---

> > > > > ### Author Response · Authors · 2025-06-06
> > > > >
> > > > > We sincerely thank you for your constructive feedback. We are committed to further refining the next version of the paper based on your suggestions.

---

> ### Author Response · Authors · 2025-06-02
> **Response to Reviewer hQ3b**
>
> ### **Q2: Effectiveness of OCR Control**
>
> We thank the reviewer for raising this important methodological discussion. We acknowledge the inherent limitations in controlling closed-source models' internal mechanisms. To validate the effectiveness of our approach, we conducted additional in-depth studies on this aspect.
>
> We conducted detailed analysis of response patterns between our baseline and *No OCR* groups, revealing behavioral differences, particularly in L5 and L7 complexity levels:
>
> | Complexity Levels | Positive Responses (baseline → No OCR) | Neutral Responses (baseline → No OCR) |
> |:------------------|:----------------------------------------:|:---------------------------------------:|
> | L5                | 80% → 50% (-30%)                         | 20% → 50%                               |
> | L7                | 84% → 6%  (-78%)                         | 16% → 94%                               |
>
> These changes demonstrate that our OCR prompt was effective: the model followed our instructions and adjusted its behavior. Under *No OCR* conditions, the model relied less on textual content and attempted to focus on visual layout patterns. However, the model still struggled to correctly interpret the visual structure, with most responses indicating that the image "lacks clear imagery to convey a strong emotional tone".
>
> We appreciate this feedback and will clarify this design choice and discuss the inherent limitations of evaluating closed-source models in our revision.
>
> ### **Q3: Handling Instruction Ambiguity**
>
> We sincerely thank the reviewer for this important clarification request. This question helps us better explain an important concept in our methodology.
>
> First, we would like to clarify that *"image sentiment"* refers to the emotional impression conveyed by the overall visual composition, not the semantic meaning of embedded text.
>
> Second, we explain why this definition is reasonable and well-grounded. Extensive psychological research demonstrates that people exhibit **emotional attention bias**—they tend to focus more on negative visual elements [7–9]. In our ASCII art, viewers first notice the negative layout, and only later recognize the non-negative characters. But due to the strong initial impression and negativity bias, the whole image still feels emotionally negative. This idea is supported by research [7–9] and also makes intuitive sense.
>
> Our study (Section 4.1) finds that VLMs often make decisions based directly on the emotion of the embedded text, without truly processing the visual layout. This is very different from how humans interpret these images.
>
> We appreciate this feedback and will expand our discussion of image sentiment definition and psychological foundations in the revised version.
>
> ### **Q4: Data and Code Release**
>
> We provide image examples and case studies in Appendix C. We will release complete materials (code and full dataset) once accepted.
>
> ### **References**
>
> [4] Wallace, E., Feng, S., Kandpal, N., Gardner, M., & Singh, S. (2019). *Universal adversarial triggers for attacking and analyzing NLP*. arXiv:1908.07125.
>
> [5] Rajpurkar, P., Jia, R., & Liang, P. (2018). *Know what you don't know: Unanswerable questions for SQuAD*. arXiv:1806.03822.
>
> [6] Chao, P., Debenedetti, E., Robey, A., Andriushchenko, M., Croce, F., Sehwag, V., ... & Wong, E. (2024). *Jailbreakbench: An open robustness benchmark for jailbreaking large language models*. arXiv:2404.01318.
>
> [7] Baumeister, R. F., Bratslavsky, E., Finkenauer, C., & Vohs, K. D. (2001). *Bad is stronger than good*. Review of General Psychology, 5(4), 323–370.
>
> [8] Öhman, A., Flykt, A., & Esteves, F. (2001). *Emotion drives attention: detecting the snake in the grass*. Journal of Experimental Psychology: General, 130(3), 466.
>
> [9] Vuilleumier, P., & Schwartz, S. (2001). *Emotional facial expressions capture attention*. Neurology, 56(2), 153–158.

---

### Decision · Program_Chairs · 2025-07-08

**Decision:**

Accept

**Comment:**

This paper proposes a controlled ASCII-art benchmark that exposes a sharp text-over-vision bias in five leading VLMs. Although two reviewers worry about scope, dataset balance and real-world impact, as well on marginal contribution with respect to previous work, the authors’ rebuttal adds balanced positive samples, an initial cross-lingual analysis, evidence of practical ASCII attacks "in the wild", and softens over-claims, leaving one clear accept, one weak accept, and two weak rejects. Given the novelty of isolating intra-modal conflict and the degradation curve presented in the paper, the remaining weaknesses (over-claims, lack of fine-tuning analysis) are not disqualifying, so I recommend a weak accept contingent on clarifying practical impact and removing overlclaiming, such as the claim that the results in this paper justify a "architectural change" in existing models. The authors are also encouraged to expand the preliminary multilingual analysis.